# Reindeer Husbandry in Switzerland—Management, Feeding, and Endoparasite Infections

**DOI:** 10.3390/ani13091444

**Published:** 2023-04-23

**Authors:** Carmen Luginbühl, Josef Gross, Christian Wenker, Stefan Hoby, Walter Basso, Patrik Zanolari

**Affiliations:** 1Clinic for Ruminants, Department of Clinical Veterinary Science, Vetsuisse-Faculty, University of Bern, Bremgartenstrasse 109a, 3012 Bern, Switzerland; patrik.zanolari@unibe.ch; 2Veterinary Physiology, Department of Clinical Research and Veterinary Public Health, Vetsuisse-Faculty, University of Bern, Bremgartenstrasse 109a, 3012 Bern, Switzerland; josef.gross@unibe.ch; 3Zoo Basel, Binningerstrasse 40, 4054 Basel, Switzerland; christian.wenker@zoobasel.ch; 4Berne Animal Park, Tierparkweg 1, 3005 Bern, Switzerland; stefan.hoby@bern.ch; 5Institute of Parasitology, Department of Infectious Diseases and Pathobiology, Vetsuisse-Faculty, University of Bern, Länggassstrasse 122, 3012 Bern, Switzerland; walter.basso@unibe.ch

**Keywords:** reindeer, husbandry, population, captivity, management, feeding, health, endoparasites, Switzerland, *Rangifer tarandus*

## Abstract

**Simple Summary:**

In Switzerland, reindeer are exclusively kept in captivity. The aim of the present work was to evaluate and summarize management and feeding practices, and to examine the prevalence of endoparasite infections in Swiss reindeer. A total of 67 animals from eight different farms and zoos were evaluated. On two visits to the farms and zoos, a standardized questionnaire was completed by the breeders and/or animal husbandry managers, the animals were weighed, and fecal samples were collected. All reindeer were fed roughage *ad libitum* and supplementary feed for reindeer or other browsers, with different compositions in each herd. The prevalence of gastrointestinal strongyles was 68.6%, with reindeer in zoos having a lower prevalence than reindeer from private farms. This study presents an overview on husbandry, feeding, and endoparasite prevalence in captive reindeer in Switzerland and provides basic data for breeders and veterinarians dealing with this deer species.

**Abstract:**

The aim of the present work was to provide an overview of management and feeding practices, and the prevalence of endoparasite infections in captive Swiss reindeer. On two visits to eight farms or zoos, a standardized questionnaire was completed. A total of 67 reindeer were weighed, and fecal samples were collected. The primary management concerns voiced by owners/managers were feeding and successful breeding. All reindeer were fed roughage *ad libitum* and supplementary feed for reindeer or other browsers, with different compositions in each herd. Males over two years of age weighed from 60 kg up to 127.5 kg, whereas females had a body weight from 53.5 kg to 86.5 kg. The prevalence of gastrointestinal strongyles was 68.6% (46/67), with reindeer in zoos having a lower prevalence (36%; 9/25) than reindeer from private farms (88%; 37/42). *Capillaria* sp., *Strongyloides* sp., and *Trichuris* sp. were detected in lower prevalences (<24%) and were also more frequent in private farms. Intestinal protozoa, as well as fluke and tapeworms, were not detected in any herd. This study provides an overview on husbandry, feeding, and endoparasite prevalence in reindeer in Switzerland and should be of help for breeders and veterinarians dealing with this animal species.

## 1. Introduction

Reindeer or caribou (Artiodactyla, Cervidae, *Rangifer tarandus*) are deer species within the genus *Rangifer* [1]. Reindeer are the only semi [2] or fully [3] domesticated deer species on a large scale worldwide, and the only deer species with a circumpolar distribution [3].

Keeping captive deer is a relatively new branch of Swiss agriculture, and only small numbers of reindeer are privately kept [4]. There is no individual registration requirement for reindeer in Switzerland. The number of reindeer must only be reported to the respective canton every two years. In Spring 2021, there were a total of 78 reindeer in Switzerland, spread across nine herds, including four herds belonging to zoos. Since 1992, captive deer have been classified as farm animals. Those who keep deer in captivity must acquire a cantonal husbandry permit and meet technical and professional qualification standards according to Swiss law. 

In Switzerland, data on the management and feeding of reindeer in captivity are scarce. Reindeer are browsers and graze on leafy plants and short leafy grasses but have a limited ability to digest long fibers rich in cellulose [2]. Depending on the climate and season, their daily food intake is approximately 2–3% of their body weight [2]. A combination of leafy forbs, seasonal green forage, or commercially available dried lichen and a pelleted feed formulated for browsers with a beet pulp base has proven adequate for reindeer in zoos [5]. In Fennoscandia, commercial reindeer feed contains different cereal grains; however, they are mainly intended for winter-feeding and therefore cover the nutritional needs of the animals during that time of the year. In North America, barley and oats are primarily used as cereal grains in reindeer pellets. The nutritional composition of feed rations for reindeer is highly variable, as some may also be produced for special purposes and adapted needs, depending on the season or category of reindeer [2].

Like other ruminants, reindeer are hosts to a variety of endoparasites that may affect their health and productivity [6]. In their natural habitat, the prevalence of subclinical, low-intensity mixed gastrointestinal parasitic infections is high [2,7,8,9]. Common gastrointestinal parasites affecting reindeer are *Eimeria* spp., gastrointestinal nematodes such as strongylids, *Capillaria* spp., *Trichuris* spp., *Skrjabinema* spp., and cestodes such as *Moniezia* spp., among others [10]. There are limited data on the prevalence of parasitic infections in captive reindeer, especially for reindeer in moderate climate zones. 

Veterinarians regularly educate clients and producers about appropriate nutrition, animal health, and preventative measures. Information on the health management of captive reindeer and the local prevalence of various endoparasites is valuable. Especially given the zoonotic potential of some animal endoparasites, such information is also important for animal caretakers, veterinarians, and human health care professionals. This is the first study on reindeer husbandry in Switzerland. 

## 2. Materials and Methods

In Spring 2021, all nine reindeer keepers (i.e., five private owners and four zoological institutions) in Switzerland were contacted to participate in this study, and eight of them accepted the invitation; one farmer declined it due to time constraints.

The participating owners or zoo veterinarians were asked to complete a questionnaire with 135 questions (Appendix A). The questionnaire included a general part with questions about the farm structure and management, and a specific part with questions about the owner’s personal interest in keeping reindeer, feeding practices, pasture use, and health status of the herd (including parasitological status, medical history including diagnosed diseases, and losses). 

In addition, individual coproparasitological analyses of all reindeer in the study were carried out between 5 September 2021 and 30 September 2021. To obtain fecal samples, animals were physically restrained for less than one minute, and rectal fecal samples were taken. The same approach was undertaken for each visit in six herds. In only two zoos were fresh fecal samples collected from the ground right after defecation. Each animal was weighed on a portable scale (TRU-TEST EziWeigh 1) and body condition score (BCS), ranging from one (low) to four (high) was assessed according to Laaksonen and Nieminen (2005) [11]. For animals that could not be captured, BCS was assessed visually, and they were not weighed. 

Fecal samples were refrigerated, transported to the laboratory, and examined within a week after collection and always by the same person. All fecal samples (*n* = 67) were examined using a combined sedimentation/flotation technique with a saturated zinc chloride solution (specific gravity 1.45) for cestode and nematode eggs, and coccidia using at least five grams of feces, and also by a sedimentation technique for trematode eggs [12]. If nematode eggs were found by the sedimentation/flotation method, a quantitative McMaster technique with a detection limit of 50 EpG (=eggs per gram feces) was performed (*n* = 44) [12]. In five of the sixty-seven samples, there was not enough material left to perform a McMaster. Fecal samples from which sufficient material was available (*n* = 53) were additionally examined for lungworms using the Baermann–Wetzel technique [12]. In one herd, a larval culture was performed. Parasite identification was performed based on morphological criteria according to Deplazes et al., 2021; van Wyck and Mayhew 2013; and Tryland et al., 2018 [2,12,13].

Aliquots for 65 of 67 fecal samples were stored at −20 °C for coproantigen detection using commercial kits. In two samples, there were not enough material left to include them in this procedure. Samples were thawed for 24 h, and then analyzed for *Giardia duodenalis* coproantigens using a commercial ELISA (MegaELISA^®^GIARDIA, MEGACOR Veterinary Diagnostics, Austria), and for *Cryptosporidium parvum* antigens using a rapid immunochromatographic test (FASTest^®^CRYTO, MEGACOR Veterinary Diagnostics, Austria). 

Microsoft Excel 365 for Windows (Microsoft Corporation, Redmond, WA, USA) was used to analyze data and to create the results. Exact binomial 95% confidence intervals were calculated with the Sample Size Calculators by UCSF Clinical and Translational Science Institute [14]. The questionnaire was analyzed by its categories. Answers were summarized and associated with the herd and its individuals. No statistical tests were performed. Prevalence was considered as the number of positive samples/total number of samples. Data subsets were created based on the age of the animals, their origin: zoo or private farm, their sex, and their parasite infections.

This study was carried out in accordance with the Swiss animal welfare legislation and was authorized by the involved cantonal animal welfare committees (approval number BE 11/2021). All participating reindeer keepers gave written consent for all examinations and for the publication of the results thereof. All collected data, as well as the subsequent evaluations, were processed confidentially.

## 3. Results

### 3.1. Reindeer Population

In autumn 2021, a total of sixty-seven reindeer were distributed on eight participating herds in Switzerland, consisting of seven Eurasian tundra reindeer (*Rangifer tarandus tarandus*) and one Eurasian wild boreal forest reindeer (*Rangifer tarandus fennicus*) herds. The average herd size was 8.15, the smallest with two animals and the biggest with nineteen. Four of the eight animal stocks were located in zoos and the rest were private keepings. The location of these herds ranged in altitude from 300 to 1300 m above sea level. The participating zoos have kept reindeer since 1935 and the private breeders since 2009 to 2018, respectively. 

The age and sex distribution of the 67 Swiss reindeer is shown in Table 1. The whole population consisted of 41 females. Half of the twenty-six male animals were infertile—nine/thirteen were castrated and four/thirteen had a temporary chemical castration. Chemical castration was achieved with a contraceptive implant containing 4.7 mg deslorelin per animal (Suprelorin^®^ ad us. vet., Virbac AG, 8152 Opfikon, Switzerland). Castrated animals only occurred in private keepings. All zoos had one adult intact male within their herd and one zoo had a male calf. All other intact males belonged to private herds (eight/thirteen).

Most animals (22/67) of the study population were between one and four years of age. Seven out of thirteen intact males were between one and four years old. Most of the males older than four years were castrated (seven/nine). There were nine calves (younger than one year). In autumn 2020, there were twenty-seven females and one male of fertilizing age. A third of the productive population reproduced in between autumn 2020 and autumn 2021.

### 3.2. Purpose of Keeping

The main reason for keeping reindeer were educational purposes (four/eight); these were all located in zoos. Other purposes for keeping were animal rental for Christmas markets, walks, photoshoots; sponsorships; or as a pure leisure pursuit. The facility with the boreal forest reindeer also aims at breeding for the conservation of the species (EAZA Ex Situ Program). None of the animal owners lived solely on the income from keeping or breeding reindeer.

### 3.3. Husbandry

All animals were kept in a permanent enclosure throughout the year. The height of the fence varied from 1.2 m up to 2.8 m. All the private keepings and one zoo had fencing of at least 2 m in height. All the herds had access to at least one stable with a concrete floor. In winter, seven of eight herds had parts of their barn floor bedded with straw or sawdust. One herd’s stable floor was lined with coffee bean shells covered with straw all year round. Four of eight herds’ enclosures comprised pasture, while the remaining four had ground with stones, gravel, or marl; three had wood chips and two a forest floor. All the herd enclosures had some hard floor with concrete at least in the barn; one herd additionally had pavement. Enclosures were cleaned once a day in six herds, twice a day in one herd, and every second day in one herd. In half of the herds (four/eight), the whole enclosure was cleaned, in three herds just the most frequented sites, and in one only the stable was cleaned. 

The artificial water facilities comprised fountains (four/eight), basins or buckets (two/eight), and automatic or self-drinkers (two/eight) in or near the barn, and a few farms had a stream or creek in the enclosure (three/eight). In winter, one herd was subjected to complete snow cover over the whole winter, three herds had occasional snow in winter, and four herds had snow only sporadically.

### 3.4. Feeding

The feeding management in each participating herd is shown in Table 2. Six of the eight herds were fed twice a day with supplementary feed. One and occasionally a second herd were fed three times a day. Only one herd was fed once a day. 

Roughage was fed *ad libitum* in all herds, but only three herd keepers could indicate the amount fed per day, so that the consumed amount could be estimated. Roughage was primarily hay of the second cut (75%), coarse hay (25%), haylage (25%), and hay chaff (13%). The coarse hay was fed together with the haylage. Five herds had the availability of grass in at least one season of the year. The feed residues were estimated by the owners or animal keepers. The estimations vary from 10% up to 50% feed residues. Most of the owners/keepers added that it varies depending on the season. 

Pelleted feed was offered in all the herds daily (see Table 2). All herds received pelleted supplementary feed formulated for reindeer or browsers (e.g., Browser 3699, Granovit, Kaiseraugst, Switzerland). The amount of supplementary feed per animal varied from 1.2 to 5.8 kg/d.

### 3.5. Animal Health and Management

The challenges in reindeer husbandry mentioned by reindeer owners/keepers more than once were feeding or changes in feeding (seven/eight); keeping males in rut, breeding season (four/eight); the management of calves (three/eight); and the management of endoparasites (three/eight). Management challenges that were mentioned once included handling wild animals; the enclosure being a stagnant pasture; the keeping of the animals within the enclosure and suitable ground; the ability of the animal to disguise clinical signs; the lack of knowledge about how a healthy animal presents; and the lack of veterinary knowledge.

In all of the privately kept herds, the reindeer were used to being haltered and were occasionally walked. In two zoos, animals were not handled, including those from which fecal samples were collected from the ground. In five of the eight herds, hoof care was performed as needed when hooves were too long, while in three herds, hoof care was not performed except when animals presented lame.

The most mentioned health problems in the questionnaire were diarrhea, lameness, and young animal problems (weak calves, sudden death of calves). Other health problems mentioned were the acute death of animals, endoparasites including two cases of *Haemonchus contortus* infection, ectoparasites, intestinal obstruction, alopecia, dental problems, babesiosis, pneumonia, enterotoxaemia caused by *Clostridium perfringens*, antler injury, emaciation, pharyngeal congestion, myiasis of the antlers, eye inflammation, and infertility. All these health problems did not necessarily result in the death of the animals.

Most diseases causing death were confirmed by a veterinarian, but not all by a pathological examination at necropsy. The most mentioned known death causes in the questionnaire were enterotoxemia, intestinal obstruction, and euthanasia because of brachygnathia inferior or because of trauma due to forkel injury (injury by antlers).

The mean body weight and range (min–max) in different sex and age groups are shown in Table 3. Fifty-six animals were weighed in six herds. Among the male calves younger than one year, the lightest calf weighed 7.5 kg and was born only two weeks before the weighing took place. In each group, weights varied by at least 20 kg between the highest and lowest weight. In the group of adult males (over two years of age), the lightest animal was less than half the weight of the heaviest. The same applies to animals younger than one year and in females that were one year old.

### 3.6. Parasitological Examinations

Gastrointestinal strongyles (GIS) were the most frequently recorded parasites both at the animal and herd levels (>50%). The prevalence of herds with *Trichuris* sp. or *Strongyloides* sp. infections was the same (two out of nine herds; 22.2%), but the prevalence of infection at the animal level was higher for *Trichuris* sp. (seven/sixty-seven; 10.4%) than for *Strongyloides* sp. (two/sixty-seven; 3%). One third of the herds had animals with *Capillaria* sp. infections, and the prevalence at the animal level was 23.9% (16/67 animals). 

Neither *Fasciola hepatica* or *Dicrocoelium dendriticum* eggs, nor larvae of any lungworm species, were detected by the sedimentation and Baermann–Wetzel techniques, respectively, in any of the 67 fecal samples. Neither *Eimeria* spp. oocysts nor tapeworm eggs were detected by the sedimentation/flotation technique. All the samples tested negative by ELISA for *Giardia duodenalis* and by the FASTest^®^ for *Cryptosporidium parvum* coproantigens.

The prevalence and the 95% confidence intervals of the detected endoparasite species in Swiss reindeer at both the animal and at the herd levels are shown in Table 4. This table does not consider mixed infections. 

The prevalence of parasite infections in the entire study population, including samples in which no parasite eggs were found and samples with mixed infections, is shown in Figure 1. It shows that 27% of the study population (*n* = 18) did not shed parasite stages or did so in such low numbers that they could not be detected. The largest proportion of the study population had a single evidenced infection with GIS (48%; *n* = 35). All the mixed infections included infection with GIS besides other parasite species. Most of them had an additional infection with *Capillaria* sp. (9%). The second most common mixed infection was a triple infection with GIS, *Capillaria* sp., and *Trichuris* sp. (6%). This was followed by a dual infection with GIS and *Trichuris* sp. (4%). Only one animal had a mixed infection with GIS and *Strongyloides* sp. (1%). In one herd, a larval culture was performed and 92% of the third-stage larvae found were identified as *Haemonchus contortus*. 

When the population was divided into two housing groups (zoos vs. private keepings), the prevalence of nematode infections was higher in privately kept animals than in zoos for all the parasites listed in Table 4 (Figure 2). For GIS infections, the confidence intervals did not overlap between zoos and private farms. However, the confidence interval overlapped for *Trichuris* sp., *Capillaria* sp., and *Strongyloides* sp. Figure 2 does not include mixed infections. 

The results of the McMaster technique, which was performed with forty-four samples derived from eight herds, are shown in Table 5. The highest GIS burden was found in a privately-owned reindeer with 13,200 EpG. The highest GIS burden found in zoo animals was 50 EpG. In five of the seven animals in which *Trichuris* sp. eggs were detected, a McMaster was performed, revealing egg burdens of 100 (*Trichuris*) EpG in one animal and <50 EpG in the four remaining reindeer. In addition, a McMaster was also performed for nine of the twelve animals infected with *Capillaria* sp., and egg counts between 50 and 400 EpG in five of the animals and <50 EpG in the remaining ones were obtained. 

Anthelmintics were used in every herd. All the zoos used the active substance fenbendazol in pellets or in a powdered form. The private herds used active substances such as fenbendazole, moxidectin, doramectin, monepantel, or a combination of levamisole and triclabendazole. In one herd, toltrazuril was administered once a year. 

## 4. Discussion

The aim of this study was to investigate the farm structure and management, feeding practices, endoparasite prevalence, and the general health situation of reindeer in Switzerland. The actual total Swiss reindeer population is small with 78 animals. Compared with other captive wild hoof stock, it corresponds to 0.6% of the fallow and red deer population (13,237 fallow and red deer) in Switzerland (https://www.sbv-usp.ch/fileadmin/sbvuspch/04_Medien/Agristat_aktuell/2021/Aktuell_AGRISTAT_2021-09.pdf) (accessed on 28 March 2023). The number of reindeer herds in Switzerland is 1.45% of registered captive deer farms (nine of six hundred and eighty-nine farms) with an average herd size of 8.15 animals and a range of two to nineteen animals. Swiss reindeer herds are smaller than captive Swiss deer herds. Although reindeer have a life expectancy of up to 20 years, the current population consisted mainly of animals less than 10 years (76%), and only one animal was over 15 years old. Most of the older animals were imported from Germany or Sweden, which provided age information at the time of import. 

The sex distribution of the Swiss reindeer population in this study is similar to that of Swiss South American camelids reported by Hengrave Burri et al. (2005) [15]. However, there are slightly fewer intact male reindeer (10%) and slightly more castrated males (13%), with an additional 6% of reindeer temporarily infertile due to chemical castration. With nine intact breeding males, there are more males than herds. However, the risk of inbreeding is high with a total population below 100 animals. This can only be solved by new imports. The similarity in sex distribution between South American camelids [15] and reindeer could be due to their similar use in private husbandry. In addition to walks, reindeer are also used for photo shoots and the rental of the animals to Christmas markets; in advertising, etc., which implies a close relationship between humans and animals; and as trained reindeer for handling. During the Christmas season, male reindeers are in rut, which makes them difficult to handle. This explains why half of the male population is castrated or temporarily infertile.

The substrate in the enclosures varied. For captive deer, a partial hard floor is recommended for good hoof abrasion, e.g., gravel or stones in the most frequented areas. However, several breeders with this type of floor have had lameness in their reindeer herd, citing sprains or sole abscesses due to entered stone. 

The feeding of the reindeer varied among herds. The dietary composition changed with the vegetation period and the seasonal availability of the feed components, whereas the composition of supplementary feed at an individual herd level largely depended on ingredient availability, preferences, and the convenience of the individual owner. Reindeer are classified as intermediate mixed ruminant feeders [16,17]. In their original habitat, they live either as semi-domesticated reindeer that still roam free on natural pastures most of the time or as wild individuals, so they feed on natural forage and migrate between summer and winter areas [2]. In winter, when feed is scarce, reindeer have the advantage to include a large portion of lichens in their diet compared with other herbivores in their natural environment. Lichens present a valuable energy source due to their high content of easily degradable carbohydrates [18]. However, only two of the participating herds in Switzerland supplemented with lichens daily because lichens either need to be collected by hand in nature or imported from northern countries. Moreover, the lichens represented only a minor dietary component in captive Swiss reindeer. Although the overall dry matter (DM) intake could not be determined in the present study, other research findings indicate that reindeer fed mainly on lichens have a lower total DM intake resulting in a loss of body mass [19,20]. When winters become harsher and an icy snow cover prevents reindeer from reaching the lichen by digging, northern reindeer owners are forced to increasingly provide their animals with supplementary feed and hay [2]. In summer, they need to compensate for the restricted nutrient intake from winter with rapid growth and the replenishment of body reserves [2]. The natural extreme environmental changes between winter and summer require an immense seasonal adaptability of the animals in their natural habitat. However, the results of this study show that their seasonal diet in Swiss captivity does not change substantially throughout the year, except for the inclusion of pasture and browse. Roughage was fed throughout the entire year. In herds with pasture access in spring and summer, the available grass of the pasture was also added to the diet during these seasons. Most of the herds (seven/eight) had access to high quality hay (second or third cut) or similar energy and nutrient rich hay as a roughage source. Reindeer are especially delicate as their digestive system cannot handle large quantities of fibers and therefore relies on roughage with a higher digestibility. Excessive crude fiber leads to the accumulation and gradual stretching of the rumen, but it remains undigested [2]. The appetite stays, but the lack of energy causes emaciation and, eventually, the death of the animal, even though the rumen is full of grass or hay [2]. However, the condition can be avoided by supplying a more easily digestible diet [2]; this is a reason why reindeer are fed additional supplementary feed to their roughage.

It is noticeable that the estimated remaining feed varied greatly between herds. This is likely caused by the estimation instead of the measurement of the remaining feed by the owners and keepers. However, constant access to roughage is crucial for ruminants, and all owners and keepers stated that roughage was fed ad libitum. 

The amount and composition of supplementary feed differed between the different Swiss herds. Zoos fed a greater amount of fresh ingredients (1.2–4.6 kg per animal per day), such as vegetables or fruit, than private farms (0–0.3 kg per animal per day). This is probably due to the better availability of vegetables and other perishable foods in zoos that are simultaneously required for other species. Furthermore, the amount of supplementary feed was higher for reindeer in zoos than in private husbandries. In general, there is a risk of over conditioning reindeer in captivity due to inadequate feeding. Reindeer’s basal metabolic rate (BMR) in winter was assumed to be 293 kJ/kg^0.75^ [21]. Especially in spring and summer, they need an energy rich diet that allows them to replenish body reserves that were mobilized during previous feed shortages. It is noteworthy that almost all herds received supplementary feed more than once a day (Table 2). For a ruminating species, such as reindeer, it is important to reduce the temporal accumulation of carbohydrates in the rumen to avoid rumen acidosis. A more frequent distribution of non-structural carbohydrate rich feed sources, such as the supplementary rations more than twice a day, would be even more desirable. In addition, most of the reported cases of intestinal obstipation occurred in one herd (herd number two) that only provided the supplementary feed once a day. The only other documented case of intestinal obstipation was because of bezoars in herd three (unpublished data). Even though the data are small, these findings indicate that it is important to spread the daily ration of supplementary feed on multiple feedings during the day, especially as pelleted feed can swell when in contact with fluids. The large amount of pelleted feed in the supplementary ration combined with stress could be a risk factor for intestinal obstructions.

Feeding was considered the most challenging aspect for owners, keepers, and zoo veterinarians when it came to dietary composition, amount, and frequency. This is illustrated through the varied feeding management approaches between farms. Reindeer must be considered a species whose dietary habits and nutritional needs may not be met through the offered feed sources available in Switzerland. Each farm or zoo chooses a different, individual approach that is based on their own empirical experience. It is important to assess the nutrients and minerals in captive reindeer and compare them with data available for samples collected from wild reindeer; this will help establish feeding recommendations for captive reindeer. 

Keeping a stag was considered difficult, especially during the rut, by at least 50% of the survey respondents. However, all herds had at least one stag or used to have one, because of breeding purposes. In 2020, of the herds that had a stag for breeding (six/eight), only three herds had live offspring, with a total of nine calves. The management of newborns and calves was also considered challenging, especially for zoos who reported that it is difficult to produce healthy offspring. When offspring is born, it is difficult to keep them alive, because they are sometimes born weak or underweight.

Of all the reported deaths, 23/41 were calves younger than six months of age. This suggests difficulties in producing live offspring and having them thrive through the first year of life. In 2020, 40% of the population were females old enough to breed and had a sexually mature stag in their herd, resulting in a total of 27 females able to reproduce. However, only nine calves were alive in autumn 2021 suggesting that one third of females were not able to reproduce successfully. Pregnancy rates in wild populations normally exceed 60% and in adult domestic flocks it may approach 100% [22]. Even though the pregnancy rate was not determined in this study, the reproduction rate seems rather small. 

Body weights varied by sex and age (Table 3). In another study, calf weights ranged from 33 to 45 kg at five to five-and-a-half months of age [21]. In this study, the mean body weight would be similar (34.9 kg for females and 34.7 kg for males) when the two youngest calves, that were born in late summer, are excluded. 

The body weights of older females ranged from 30.5 kg to 86.5 kg, with females weighing less than 60 kg in all age groups. The optimal criteria for reproduction and calf production cannot be met if the live weight of the females does not exceed 60 kg [23,24]. It is important to further investigate whether the low body weights may have caused the poor calving rate, especially since the animals from two zoos that had problems with reproduction could not be weighed. The body weights of male animals also varied substantially when compared with three-year-old males that weigh between 88 kg and 97 kg [25]. However, the number of animals weighed in this study was small and must be interpreted with caution. For future studies concerning calving rates, we recommend weighing cows prior to the rutting season to then compare with the reproductive success the following year. Furthermore, the heavy burden of parasites should also be considered if fertility is low. Other reasons, such as management factors regarding herd size, herd composition, or bulls, should also be considered. 

Reindeer can harbor a variety of endoparasites. The prevalence of endoparasite infections in this study population (*n* = 67) was 73% or 49/67 (Figure 1). Although protozoa such as *Cryptosporidium* spp., *Giardia* duodenalis or, or *Eimeria* spp. may infect reindeer [9], we did not find any of these protozoa in our study. This is in contrast to caribou from Northern Alaska where the overall prevalence of *Cryptosporidium* oocysts was 6.1% in [26]. On the other hand, in Northern Norway, no *Cryptosporidium* oocysts were found [6]. Cryptosporidiosis, especially *C. parvum* infection, is a common diarrheic disease in bovine calves, and the infection can also be transmitted to humans and other animal species. 

*Giardia duodenalis* was not detected in any of the samples from this study, which is in agreement with the findings in caribou from Northern Alaska [26]. In contrast, in 2021, a prevalence of 5% of *G*. *duodenalis* cysts was detected in reindeer from Northern Norway [6]. *Giardia duodenalis* has a high prevalence in various species of young animals and it may be a cause of diarrhea [12,27]. Although only *G. duodenalis* would be expected to be present in reindeer, different *Cryptosporidium* species have been described in this deer species [28,29]. The commercial tests used in this study were designed to detect coproantigens of *G. duodenalis* and *C. parvum*, but it is not known if the used commercial kit would be able to detect coproantigens from other *Cryptosporidium* species described in reindeer besides *C. parvum*.

Intestinal coccidia species are usually specific to their host species. *Eimeria* spp. are also frequently found in semi-domesticated reindeer calves, whereas the prevalence in adult wild reindeer is low [9]. Nevertheless, the prevalence of *Eimeria* spp. was 51% in reindeer calves in 2019 on Fennoscandia with the higher density of reindeer considered a risk factor for shedding oocysts [10]. Interestingly, there were no *Eimeria* spp. or *Isospora* spp. found in this study. This was unexpected, especially when considering the higher density of captive animals compared with wild populations. Possible reasons are the small number of calves distributed in only three herds in 2021, or the import of reindeer into Switzerland who were free of intestinal coccidia and, therefore, no infection source for young animals was present in this country.

Compared with protozoa, more gastrointestinal nematodes were found in the Swiss reindeer population. The highest prevalence of 68.9% (CI 95%: 56.2–79.4) was identified for GIS. This prevalence is very similar to the prevalence of 75.6% reported in 2019 for reindeer calves in Fennoscandia [9]. An abomasal nematode reported in reindeer is *Haemonchus contortus* [30]. *Haemonchus contortus* infections result in blood loss and consequent anemia. Cases of *H. contortus* infections have been reported in reindeer in Switzerland (C. Luginbühl personal communication). In addition, coprocultures and the differentiation of third-stage larvae were performed on one farm during this study, confirming a heavy burden of *H. contortus* in all animal age groups. The findings from this farm were also confirmed by the Swiss Consulting and Health Service for Small Ruminants (Niederönz, Bern, Switzerland) (unpublished data). The clinical signs of haemonchosis in the farm were diarrhea, emaciation, pale mucous membranes, and submandibular edema in a late stage. In this same herd, the highest EpG numbers were found. The animal, which was hardest hit by clinical signs had a count of 7100 EpG in our fecal analyses on samples taken one week before the clinical signs were strongest. The herd was treated with levamisol and triclabendazol and the animal fully recovered. A McMaster performed two weeks after deworming showed an almost 100% efficacy of the anthelminthic treatment. 

Another parasite of ruminants that occurs in the small intestine is *Capillaria* sp. In this study, the prevalence of *Capillaria* sp. was 23.9% (CI 95% 14.3–35.9), which was lower than in other studies that found a prevalence of 60% in semi-domesticated reindeer calves, while the frequency was much lower in their adult counterparts [31]. In addition to being a frequent parasite in reindeer calves, *Capillaria* spp. appear to be a predominant parasite in reindeer zoo enclosures [32]. It is interesting to note that only one of the three Swiss herds with *Capillaria* sp. infections was a zoo. Two of these herds, including the zoo, had calves in 2021. Although these parasites shed eggs all year-round, they appear to be most prevalent during the cold winter months [9], which may be one reason why prevalence was not higher in Switzerland, where samples were collected in early fall. *Capillaria* sp. has also not been associated with specific clinical signs in ruminants [33], yet it was documented that severe infection can lead to enteritis and diarrhea [34]. 

Commonly known as “whipworms”, Trichuridae is another parasite family that colonizes the intestines of many mammalian species. Infection with *Trichuris* spp. can lead to acute or chronic inflammation in the caecum [9]. Although *Trichuris* spp. are not commonly observed in reindeer in Europe [35,36], at least five *Trichuris* spp. have been reported from Russian reindeer herds [37]. The results of this study showed a prevalence of 10.4% (CI 95% 4.3–20.4) in Swiss reindeer. Compared with a study in Fennoscandia, where the prevalence of *Trichuris* sp. in reindeer calves was only 0.6%, the prevalence in Switzerland seems to be high [9]. In wild moose, infections with these parasites are not considered a threat [35], but they can cause bloody diarrhea, especially in young animals [38]. Furthermore, it is thought that *Trichuris* spp. may be a component of the Wasting Syndrome Complex, a condition involving chronic diarrhea and the loss of body weight and condition [39,40]. There is also a report of a reindeer’s death, which was infected with “whipworms”, in a Finnish zoo [9]. However, Figure 1 shows that all animals with a *Trichuris* sp. infection had a co-infection with GIS or GIS and *Capillaria* sp. Furthermore, the only two herds where animals shed *Trichuris* sp. eggs were herds that had calves. Five out of the seven *Trichuris* infections were recorded in calves younger than one year old. The other two animals were one-and-half and eleven-and-a-half years old. It is evident that these two animals with 13,200 EpG and 10,750 EpG excreted the most GIS eggs per gram of feces of the entire study population (Table 5). 

The difference in prevalence of parasite infection in zoos and in private farms is shown in Figure 2. Why reindeer in zoos seem to be less frequently infected with nematodes is unclear. However, zoos have fewer calves (three/nine) and thus fewer reservoirs, but the number of calves in the entire study population was small, and the only zoo with calves had no infections with GIS, only some infections with *Capillaria* sp. in its calves. Another reason could be that reindeer in private keepings have more pastures in their enclosure, from which they also feed. These pastures could be an ideal place for the parasites to complete their life cycle and (re-)infect their hosts. Reindeer without pastures may spend less time feeding from the ground, where they also defecate, and therefore have less chance of (re-)infecting themselves. Another argument could be that more areas, sometimes even the entire enclosure, are cleaned more frequently in zoos than in private keepings resulting in a reduced parasite burden. 

## 5. Conclusions

This study is the first to evaluate captive reindeer husbandry, feeding, and management practices in Switzerland. The findings that GIS are more common in some herds will help reindeer keepers and veterinarians to better plan diagnostic and preventative strategies, especially concerning management practices. However, it would be of value to further investigate this subject over a longer period of time to see how parasite burden changes throughout the year and to focus on management factors to determine which are crucial. 

## Figures and Tables

**Figure 1 animals-13-01444-f001:**
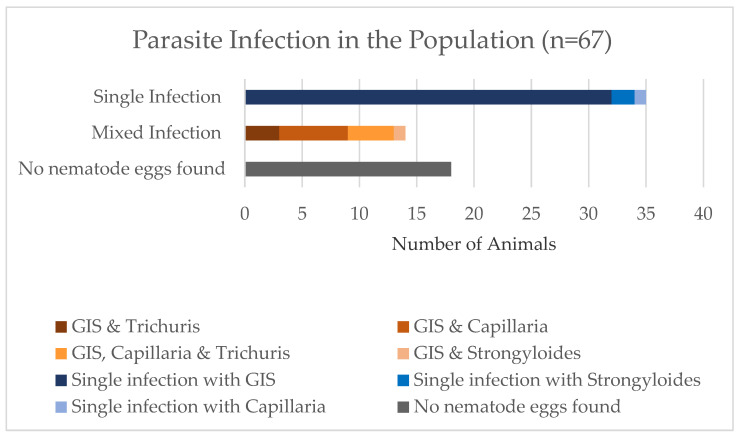
Parasite infections in the Swiss reindeer population (*n* = 67). Mixed infections are considered.

**Figure 2 animals-13-01444-f002:**
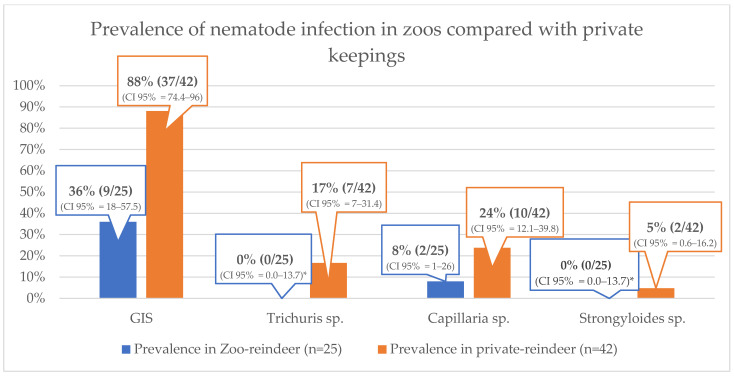
Prevalence of nematode infections inn zoo-reindeer compared with reindeer in private keepings (* one-sided 97.5% confidence interval).

**Table 1 animals-13-01444-t001:** Age and sex distribution of Swiss reindeer in eight herds (*n* = 67).

Age	Females	Males			Age Distribution
		Intact	Castrated	Temporary Infertility with Chemical Castration	
<1 year	5	4	0	0	9
1–4 years	11	7	1	3	22
>4–10 years	11	2	6	1	20
>10 years	14	0	2	0	16
Sex distribution	41	13	9	4	

**Table 2 animals-13-01444-t002:** Feeding practices and estimated amount of feed intake in captive reindeer per animal (in kg per day) calculated based on the daily ration given to the herd. No differences were made between female, males, or calves.

	Herd 1	Herd 2	Herd 3	Herd 4	Herd 5	Herd 6	Herd 7	Herd 8
Numbers of feedings per day	2	1	2	2	2 to 3	2	2	3
Roughage								
Coarse hay	unknown		unknown					
Second/Third cut hay		0.31	unknown	unknown	2.15	unknown		unknown
Haylage	unknown		unknown					
Hay chaff							>1.33	
Fresh cut herbage/pasture	pasture	pasture		pasture			in spring	in spring
Browse (*Salix* sp.)					spring to autumn			
Estimated feed residues	30%	unknown	10%	<50%	15%	unknown	30%	20%
Supplementary feed								
Granovit 3699 ^1^	0.5				1.2	0.7	0.7	1.1
Reindeer feed Mühle Bachmann ^2^		0.8	0.4					
Reindeer feed Mühle Strahm ^3^				1.1				
Corn pellets	0.4	1		0.9	0.1			
Lucerne	0.15					0.2		
Sugar beet pellets								0.1
Wheat bran	0.1							
Raw fibre cubes ^4^			0.45					
Beet pulp pelleted					0.25			
Lichen		0.02			1.26			
Vegetables			0.3		2.32	in winter	4.6	2.2
Fruits					0.65			
Cod liver oil and bran						1.2		
Olive oil			0.003					
Bread						0.05		
Mineral Feed								
Megaflor ^5^			0.03					
Cattle salt			0.015					
Totalin ^6^			0.001					
Mineral mixture						0.032		
Mineravit ^7^							0.01	
Supplementary and mineral feed in total (kg/animal/day)	1.2	1.8	1.2	2.0	5.8	2.2	5.3	3.4

^1^ Granovit 3699 ingredients: lucerne meal, apple pomace, sunflower meal, linseed products, sugar beet pulp, potato protein, cellulose vitamin, and mineral premix including bicarbonate, molasses, corn Reindeer feed; ^2^ Mühle Bachmann ingredients: wheat, corn, barley, oats, soybean meal, lucerne meal, grain chaff, dried sugar beet pulp, corn gluten meal, trace elements + vitamins, carob, sugar beet molasses, monocalcium phosphate, carbonic acid algal lime, magnesium oxide, calcium carbonate; ^3^ Reindeer feed Mühle Strahm ingredients: wheat, corn, oats, barley, sunflower cake, lucerne meal, soybean meal, minerals, corn gluten meal, apple pomace, carob, dried sugar beet pulp, sugar beet molasses, vegetable oil (soy), selenium Vit. E; ^4^ Raw fibre cubes, Mühle Bachmann AG, Ingredients: spelt husks, soybean extraction meal, alfalfa cubes, sunflower cake, raw cellulose, biscuit meal, vitamins and minerals, corn gluten, carob germ and hull meal, apple pomace, dried sugar beet pulp, sugar beet molasses, vegetable oil (soy); ^5^ Megaflor, Anitech, Ingredients: wheat bran, cod liver oil, linseed cake, rapeseed cake, sunflower cake, oats, dicalcium phosphate, wheat germ, sodium chloride, trace elements and vitamins; ^6^ Totalin, Biokema SA, Ingredients: tricalcium phosphates, pentahydric copper sulfates, black iron oxide, iodine, heptahydric magnesium sulfates, manganese (II) monohydric sulfates, sodium chloride, dihydrogen sodium phosphates, monohydric zinc sulfates, dried de-enzymatic yeast, fennel fruit, minerals, retinol-acetate/palmitas, cholecalciferol, alpha-tocopherol acetate, seaweed powder; ^7^ Mineravit, Dr. E. Graeub AG, Ingredients: minerals, cereals, carob flour, fennel, trace elements, vitamins, flavorings.

**Table 3 animals-13-01444-t003:** Body weight of captive reindeer by age and sex (*n* = 56).

Age	Females		Males	
	Animals Weighed	Mean (Min; Max) in kg	Animals Weighed	Mean (Min; Max) in kg
<1 year	5	30.9 (15; 53.5)	4	27.9 (7.5; 42)
1 year	3	46.2 (30.5; 62)	3	63 (54.5; 78)
2 years	4	57.9 (46; 66)	3	82.2 (70; 90)
>2 years	20	70.1 (53.5; 86.5)	14	100.3 (60; 127.5)

**Table 4 animals-13-01444-t004:** Prevalence of parasite species in Swiss reindeer (*n* = 67) and reindeer herds (*n* = 8). Mixed infections are not considered.

Parasite Species	Positive/Tested Animals	Prevalence% (95% CI) in Animals	Positive/Tested Herds	Prevalence% (95% CI) in Herds
Gastrointestinal strongyles (GIS)	46/67	68.6 (56.2–79.4)	7/8	87.5 (47.4–99.7)
*Capillaria* sp.	16/67	23.9 (14.3–35.9)	3/8	37.5 (8.5–75.5)
*Trichuris* sp.	7/67	10.4 (4.3–20.4)	2/8	25.0 (3.2–65.1)
*Strongyloides* sp.	2/67	3.0 (0.4–10.4)	2/8	25.0 (3.2–65.1)

**Table 5 animals-13-01444-t005:** Number of Gastrointestinal strongyles (GIS) eggs per gram feces (EpG) (*n* = 44).

Husbandry System	Tested Animals	Lowest GIS EpG	Highest GIS EpG	Mean of GIS EpG	Median Value of GIS EpG
Private farms (*n* = 42)	35	0	13,200	1543	400
Zoos (*n* = 25)	9	0	50	11	0

## Data Availability

The datasets generated and/or analyzed during the current study are not publicly available due to the individual privacy of Reindeer owners but are available from the corresponding author on reasonable request.

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
