# Peer review of "Reindeer Husbandry in Switzerland—Management, Feeding, and Endoparasite Infections"

_animals, 2023, doi:10.3390/ani13091444_

Round 1

Reviewer 1 Report

Manuscript ID: animals-2341725

Title: Reindeer Husbandry in Switzerland – Management, Feeding 2 and Endoparasite Infections

 Authors: Carmen Luginbühl, Josef Gross, Christian Wenker, Stefan Hoby, Walter Basso, and Patrik Zanolari

 General Comments:

The authors of this manuscript used a 135-question survey of reindeer owners/managers to evaluate many animal husbandry practices for 8 of the 9 reindeer herds in Switzerland and also weighed most animals and performed a very standard fecal analysis for endoparasites in most reindeer from these 8 herds. Four of the herds were from zoos and 4 were from private owners. The survey questions fit into 5 categories: 1) general questions related to the physical facilities, 2) questions specific to their enclosed structure(s), 3) questions about feeding practices, 4) questions related to herd management, 5) questions about any herd health issues, including deworming practices. While concerns were raised by the owners/managers about the low reproductive successes and poor calf outcomes in their herds, there were no questions specifically related to calving rates. Body weight data was consistent with expectations related to gender and age groups, but mature female weights were thought to be a little low. The parasitology methods were generally appropriate for the expected endoparasites and for reindeer. The commercial kits used to detect Giardia and Cryptosporidium  were designed for the zoonotic species of each (i.e. G. duodenalis and C. parvum), and so they may miss some of the other potential species/subspecies of each. Prevalence levels for the nematode eggs identified in this study were typical of livestock raised under these conditions. The presence of small numbers of trichostrongyles, Trichuris sp., Capillaria sp. and Strongyloides sp. cause few if any health issues, and so reporting prevalence levels provide little practical value for assessing potential health issues from these nematodes without knowing the intensity levels. Only the trichostrongyle nematode eggs were quantified, but the other nematode eggs should have also been quantified even if intensities were found to be below detectable levels with the McMaster method. Trichostrongyle intensities appeared to be high in at least one of the reindeer from a private herd, but very little intensity data was given. It would be useful to report out the individual herd data and potentially relate this to management practices in any herds with especially high levels. The survey contained several questions related to anthelmintic uses within the herds, and so it was surprising that there was no mention of these results in the manuscript. The absence of fluke and tapeworm eggs is noteworthy but not surprising until these husbandry conditions, but the absence of coccidian oocysts in all of the 67 reindeer is quite surprising. The authors speculated on some potential reasons for this absence; the potential use of chemotherapeutic preventatives for coccidiosis was not included. Questions about their use was not included in the survey, but it could be excluded as a reason by contacting 8 owners/managers.     

This is a well-organized and well-written manuscript; however, there are several instances where minor grammatical corrections need to be made. While the experimental designs for this study were solid, the only novelty of this study is it’s location and the very low number of reindeer present throughout Switzerland. In their current form, results from this study have only a regional value. While the quality of the study and the composition of the manuscript is worthy of publication, the authors should consider ways to increase its value beyond a very baseline evaluation of 8 herds. For example, one approach could be to provide recommendations for future studies, particularly looking at how to improve calving rates among these herds, but also in isolated populations of reindeer in other regions. The authors use considerable space speculating about all the potential trichostrongyle species that might be present in herds and the coccidian species that weren’t in the herds. This information provides little value to the manuscript and could be substituted with recommendations for futures studies that would have greater value. Other comments related to specific sections of the manuscript are found below.

Title:

1.  The title provides a good vison for the content of the manuscript.

Simple Summary

1.    This summary provides a good overview of the objectives and findings of this study.

2.    The terminology used is appropriate for a readers outside of the discipline and should be understandable for livestock producers.

Abstract:

1.    The abstract is clear and contains the appropriate level of detail.

2.    The absence of any protozoan parasites is mentioned but the absence of fluke and tapeworm eggs should also be included.

3.    If space allows, I believe that it would be helpful to include the primary management concern voiced by the owners/managers.

Keywords:

1.    It would seem that the country name “Switzerland” would be used instead of “Swiss”

Introduction:

1.    This section is appropriate for the study.

2.    While “data on management and feeding of reindeer in captivity are scarce” in Switzerland, there is a wealth of resources available from other regions. It would be useful to producers to recommend at least one of these good general husbandry resources in the introduction.

Methods:

1.    Parasite levels can vary significantly depending on the time of year. The study was started in September, and so its possible that all fecal collections were collected soon after that. Yet, dates (at least rough estimates) for the fecal collections should be specified.

2.    Listing the Excel package used for data calculations is not needed for this section.

3.    The description for the data analyses of the questionnaire is too vague. If any statistics were  involved, this should be specified.

4.     

Results:

1.    The one boreal forest reindeer herd is identified to the subspecies level (Rangifer tarandus fennicus), but not the subspecies of the other 7 herds. I assume that they are Rangifer tarandus tarandus based upon its widespread distribution, but it should be specified.

2.    The age and sex distributions were adequately described and so it should be possible to calculate calving rates if the number of calves born is known. Even though the sample size might be low, this baseline data might become useful.

3.    The data available on trichostrongyle intensities should be expanded to each herd and relate this information to management style. Were there herds that had intensity levels significantly higher than others? Presumably, these were the pastured herds that were not routinely dewormed, but this should be specified.

4.    Anthelmintic use data was collected, but not reported in the manuscript. Is there a reason for this? What about the use of anticoccidials? Given the absence of any coccidian oocysts, it would be good to verify that these products were not used.

5.    If available, intensity levels should be mentioned for the other nematodes. Prevalence levels were low but were there any herds with high intensities.

6.    The absence of Fasciola hepatica or Dicrocoelium dendriticum eggs nor lungworm larvae is reported in the results section, but nothing is said about tapeworm eggs. It might be assumed that they are also absent, but it should be specified.

Discussion:

1.    The information needed to calculate calving rates is included in the discussion, and it appears to be two-thirds of the bred females for 2020. This seems to be an important issue for owners/managers and so this should be presented clearly in the result section and expanded to other years if possible. In the discussion, the authors should explain if these values are low for similar small reindeer herds or if this is expected considering the conditions. The suggestion that this might be related to slightly lower weights for adult females is fine, but should be better supported with the weight data (if possible). This issue could be developed more as part of a section of recommended future studies.

2.    The discussion about coccidian was relatively concise and included the necessary information considering that oocysts were not found. It still could be shortened even more without eliminating critical information. The list of possible explanations for the absence of oocysts could also include the frequent removal of manure from pens, and possibility of owners/managers using chemotherapeutic preventatives for coccidiosis. The latter could be eliminated by contacting the 8 owners/managers and asking them this question.

3.    The results of larval cultures from the one farm with heavy H. contortus burdens in all age groups of animals must be first reported in the results section. This is a very important finding a should be described more fully. Why was this herd singled out? Was this the herd that contain fecal egg counts above 13,000? Were these animals showing clinical signs of haemonchosis? What was the percentage of Haemonchus larvae? What was the grazing conditions of the herd? Reporting prevalence levels for these common parasites provides little information on the relative importance of these nematodes, but finding a clinical problem demonstrates that this type of problem is possible under these conditions. Therefore, it would be useful to describe the appropriate management factors used in this herd. Can anemia be easily detected in reindeer that are routinely handled?

4.    Since the other trichostrongyles were not specifically identified any of the herds, it seems unnecessary to provide detailed descriptions on the possible species. The important information could be transmitted with a simply reference, and this would save about a page worth of text. The low prevalence rates for the other nematodes is typical for many other ruminants, and at these levels they are not considered to be important. The general information described for each of these other nematodes could also be shorted considerably.  

Conclusions:

1.    The authors conclude that. “The findings that GIS are more common in private farms will help reindeer keepers and veterinarians to better plan diagnostic and preventative strategies. While this statement is true, it also points out the weakness of just comparing privately-owned to zoo-owned herds. Type of ownership would not be a factor in the presence and/or intensity of parasitism. For the parasitology portion of this study to have any value, the authors must report parasite intensities for each herd and then  compare them to actual management factors. Even with the data that has been already collected, it should be possible to do this with the trichostrongyles.

Author Response

Response to Reviewer 1 Comments

Point 1: While concerns were raised by the owners/managers about the low reproductive successes and poor calf outcomes in their herds, there were no questions specifically related to calving rates.

Response 1: We agree with the reviewer that there were no questions asked specifically related to calving rates. This is because the subject was not supposed to be of greatimportance for this review. Also the problem of low reproductive success and poor calf outcomes in some herds was not identified until the questionnaire itself was completed.

Point 2: The commercial kits used to detect Giardia and Cryptosporidium were designed for the zoonotic species of each (i.e. G. duodenalis and C. parvum), and so they may miss some of the other potential species/subspecies of each.

Response 2: Additions were made in the main text in the discussion.

Point 3: Prevalence levels for the nematode eggs identified in this study were typical of livestock raised under these conditions. The presence of small numbers of trichostrongyles, Trichuris sp., Capillaria sp. and Strongyloides sp. cause few if any health issues, and so reporting prevalence levels provide little practical value for assessing potential health issues from these nematodes without knowing the intensity levels. Only the trichostrongyle nematode eggs were quantified, but the other nematode eggs should have also been quantified even if intensities were found to be below detectable levels with the McMaster method.

Response 3: We agree with the reviewer and appreciate the notification. We added the results of McMaster for Trichuris sp. and Capillaria sp. to the results.

Point 4: Trichostrongyle intensities appeared to be high in at least one of the reindeer from a private herd, but very little intensity data was given. It would be useful to report out the individual herd data and potentially relate this to management practices in any herds with especially high levels.

Response 4: We agree with the reviewer, however this was intentionally omitted so as not to provide too much information about a particular herd in order to preserve the owner/manager's rights to privacy, especially since the study population is so small and there are only a few herds in Switzerland. We do not want to create a disadvantage for owners/managers by having participated in the study.

Point 5: The absence of fluke and tapeworm eggs is noteworthy but not surprising until these husbandry conditions, but the absence of coccidian oocysts in all of the 67 reindeer is quite surprising. The authors speculated on some potential reasons for this absence; the potential use of chemotherapeutic preventatives for coccidiosis was not included. Questions about their use was not included in the survey, but it could be excluded as a reason by contacting 8 owners/managers.

Response 5: One herd reported in the questionnaire, that Toltrazuril is used once a year. The other herd owners/managers were contacted and asked.

Point 6: This is a well-organized and well-written manuscript; however, there are several instances where minor grammatical corrections need to be made. While the experimental designs for this study were solid, the only novelty of this study is it’s location and the very low number of reindeer present throughout Switzerland. In their current form, results from this study have only a regional value. While the quality of the study and the composition of the manuscript is worthy of publication, the authors should consider ways to increase its value beyond a very baseline evaluation of 8 herds. For example, one approach could be to provide recommendations for future studies, particularly looking at how to improve calving rates among these herds, but also in isolated populations of reindeer in other regions.

Response 6: Thanks to the reviewer for this helpful remark. We added our thoughts concerning future studies to the text. However, it is difficult to give special advice for improving calving rates with just this study, because numbers of females and bulls supposed to breed were small, as well as the offspring numbers the following year. Furthermore, the study is only a “snapshot” of the reindeer husbandry and breeding in Switzerland.

Point 7: The authors use considerable space speculating about all the potential trichostrongyle species that might be present in herds and the coccidian species that weren’t in the herds. This information provides little value to the manuscript and could be substituted with recommendations for futures studies that would have greater value.

Response 7: We thank the reviewer for pointing this out. We made adjustments in the text.

Point 8 (Abstract): The absence of any protozoan parasites is mentioned but the absence of fluke and tapeworm eggs should also be included.

Response 8: We thank the reviewer for this addition. We added it in the abstract.

Point 9 (Abstract): If space allows, I believe that it would be helpful to include the primary management concern voiced by the owners/managers.

Response 9: We agree, we added it in the abstract.

Point 10 (Keywords): It would seem that the country name “Switzerland” would be used instead of “Swiss”

Response 10: We thank the reviewer for pointing this out. We changed the keyword in the manuscript.

Point 11 (Introduction): While “data on management and feeding of reindeer in captivity are scarce” in Switzerland, there is a wealth of resources available from other regions. It would be useful to producers to recommend at least one of these good general husbandry resources in the introduction.

Response 11: We agree with the reviewer and made additions in the main text.

Point 12 (Methods): Parasite levels can vary significantly depending on the time of year. The study was started in September, and so its possible that all fecal collections were collected soon after that. Yet, dates (at least rough estimates) for the fecal collections should be specified.

Response 12: We thank the reviewer for pointing this out. Adjustments were made in the main text.

Point 13 (Methods): Listing the Excel package used for data calculations is not needed for this section.

Response 13: Adjustments were made in the main text.

Point 14 (Methods): The description for the data analyses of the questionnaire is too vague. If any statistics were  involved, this should be specified.

Response 14: We thank the reviewer for this point of view. We made the adjustments in the text.

Point 15 (Results):  The one boreal forest reindeer herd is identified to the subspecies level (Rangifer tarandus fennicus), but not the subspecies of the other 7 herds. I assume that they are Rangifer tarandus tarandus based upon its widespread distribution, but it should be specified.

Response 15: We thank the reviewer for pointing this out. We made the adjustments in the main text.

Point 16 (Results): The age and sex distributions were adequately described and so it should be possible to calculate calving rates if the number of calves born is known. Even though the sample size might be low, this baseline data might become useful.

Response 16: We agree with the reviewer. We have also included calculations for the year 2020 to 2021 in the results. However, further calculations for other years are difficult because some herds were split up during breeding season and another herd had males that were supposed to be chemically castrated, but still had an offspring the following year. As one might see, despite the small number of animals , the control of breeding and producing offspring is not as one would like for the analyses.

Point 17 (Results): The data available on trichostrongyle intensities should be expanded to each herd and relate this information to management style. Were there herds that had intensity levels significantly higher than others? Presumably, these were the pastured herds that were not routinely dewormed, but this should be specified.

Response 17: We understand the reviewers point of view, however, but did not want to publish too much data on individual herds, as this could be a disadvantage for them. Especially because the intensity of GIS was higher in private farms and these herds that would suffer the most from other owners knowing their parasite burden.

Point 18 (Results): Anthelmintic use data was collected, but not reported in the manuscript. Is there a reason for this? What about the use of anticoccidials? Given the absence of any coccidian oocysts, it would be good to verify that these products were not used.

Response 18: We thank the reviewer for this comment. We have added these missing results. However, exact dosages can not be given because we were not the reating veterinarian.

Point 19 (Results): If available, intensity levels should be mentioned for the other nematodes. Prevalence levels were low but were there any herds with high intensities.

Response 19: We added these results.

Point 20 (Results): The absence of Fasciola hepatica or Dicrocoelium dendriticum eggs nor lungworm larvae is reported in the results section, but nothing is said about tapeworm eggs. It might be assumed that they are also absent, but it should be specified.

Response 20: We thank the reviewer for the information, it was added in the text.

Point 21 (Discussion): The information needed to calculate calving rates is included in the discussion, and it appears to be two-thirds of the bred females for 2020. This seems to be an important issue for owners/managers and so this should be presented clearly in the result section and expanded to other years if possible. In the discussion, the authors should explain if these values are low for similar small reindeer herds or if this is expected considering the conditions. The suggestion that this might be related to slightly lower weights for adult females is fine, but should be better supported with the weight data (if possible). This issue could be developed more as part of a section of recommended future studies.

Response 21: Unfortunately, it is not possible to support the assumption regarding the slightly lower weights with the weight data because there where even less calves the next year (2022) when females were weighed in autumn 2021, and there were no data about the weight of females in 2020. However, we made some adjustments to compare the calving rate with pregnancy rates.

Point 22 (Discussion): The discussion about coccidian was relatively concise and included the necessary information considering that oocysts were not found. It still could be shortened even more without eliminating critical information. The list of possible explanations for the absence of oocysts could also include the frequent removal of manure from pens, and possibility of owners/managers using chemotherapeutic preventatives for coccidiosis. The latter could be eliminated by contacting the 8 owners/managers and asking them this question.

Response 22: We chose not to include manure removal from the pens as an additional reason, because two-third of the calves were kept in private herds where the entire pen is not cleaned. These herds would also be expected to have coccidia, as the calves may be a reservoir, and in addition, these two herds were found to have the highest parasite loads of other parasite species such as trichostrongyles, Trichuris spp. and Capillaria spp.

Furthermore, it is not common practice in Switzerland to use preventatives in farm animals like reindeer such as chemotherapeutic preventatives in rabbit feed. Only one farm mentioned the use of Toltrazuril.

Point 23 (Discussion): The results of larval cultures from the one farm with heavy H. contortus burdens in all age groups of animals must be first reported in the results section. This is a very important finding a should be described more fully. Why was this herd singled out? Was this the herd that contain fecal egg counts above 13,000? Were these animals showing clinical signs of haemonchosis? What was the percentage of Haemonchus larvae? What was the grazing conditions of the herd? Reporting prevalence levels for these common parasites provides little information on the relative importance of these nematodes, but finding a clinical problem demonstrates that this type of problem is possible under these conditions. Therefore, it would be useful to describe the appropriate management factors used in this herd. Can anemia be easily detected in reindeer that are routinely handled?

Response 23: We agree with the reviewer. We signposted the herd/case because we have tried to maintain the privacy of the herds, especially because there are so small numbers of reindeer owners in Switzerland that it would be easy for people to find out which cases belong to which herd.

Point 24 (Discussion): Since the other trichostrongyles were not specifically identified any of the herds, it seems unnecessary to provide detailed descriptions on the possible species. The important information could be transmitted with a simply reference, and this would save about a page worth of text. The low prevalence rates for the other nematodes is typical for many other ruminants, and at these levels they are not considered to be important. The general information described for each of these other nematodes could also be shorted considerably.

Response 24: We agree with the reviewer and made the removed unnecessary information from the manuskript.

Point 25 (Conclusion): The authors conclude that. “The findings that GIS are more common in private farms will help reindeer keepers and veterinarians to better plan diagnostic and preventative strategies. While this statement is true, it also points out the weakness of just comparing privately-owned to zoo-owned herds. Type of ownership would not be a factor in the presence and/or intensity of parasitism. For the parasitology portion of this study to have any value, the authors must report parasite intensities for each herd and then  compare them to actual management factors. Even with the data that has been already collected, it should be possible to do this with the trichostrongyles.

Response 25: We thank the reviewer for the helpful remarks and advice. We made adjustments in the conclusion.

Reviewer 2 Report

I consider the submitted manuscript as very timely and informative contribution to the emerging problem – scientific basis to ensure proper keeping of reindeers in the areas outside their modern (!) areal of distribution. Authors describe the reasons, why reindeers are kept in captivity in such unexpected places as Swiss Confederation, and with some surprise we can see, that the same reasons (leisure industry) are true in other countries. In our experience, we can see that reindeers are kept for New Year festivities, as mini-zoos or as components of ‘ethnic parks’. We also witness lower level of care and general conditions of herd keeping in some private owned husbandries. Very often, the parasitological status of animals in such minor herds is unacceptably low.

               Submitted manuscript demonstrates one very positive quality: observations on the technological aspects of reindeer keeping (general management and feeding practices) are combined with the pathological and parasitological data. I am sure, that such a paper will be very interesting for all the people involved into reindeer keeping and can even represent an example of such multi-faceted approach to the problem. The paper is ready for publication and we would like to attract the attention of authors to some minor editorial moments or misprints.

               The main methodological moments of the paper are described in full, but the approach to the identification of parasites is not obvious from M&M. I guess that just the reference on one or two sources of information (books, web sites etc.) used to identify the parasites and their stages will be enough. Such reference will be also useful for the ‘followers’ – reindeer keepers which will decide to undertake similar studies in other places. Despite the broad application of molecular techniques for parasites identification in my personal opinion, such approach is not needed in the case of this study. Morphological approach is sufficient for the goals of this survey.

I am not native speaker of English and some expressions of the submitted paper were not clear for me.

E.g. authors use the expression ‘opportunistic browser’ or ‘serial browse’. All my attempts to use common dictionaries to disclose the sense of such usage of  this word were not successful. Probably some explanation can be inserted at the first use of the term.

               Probably authors will re-consider the use of commercial names as ‘Tricalcii phosphas’ (page 6) in favor IUPAC names of chemical compounds – in this case ‘tricalciumphosphat’?

Author Response

Response to Reviewer 2 Comments

Point 1: The main methodological moments of the paper are described in full, but the approach to the identification of parasites is not obvious from M&M. I guess that just the reference on one or two sources of information (books, web sites etc.) used to identify the parasites and their stages will be enough. Such reference will be also useful for the ‘followers’ – reindeer keepers which will decide to undertake similar studies in other places. Despite the broad application of molecular techniques for parasites identification in my personal opinion, such approach is not needed in the case of this study. Morphological approach is sufficient for the goals of this survey.

Response 1: We appreciate this remark, we made the adjustments in the manuscript.

Point 2: I am not native speaker of English and some expressions of the submitted paper were not clear for me.

E.g. authors use the expression ‘opportunistic browser’ or ‘serial browse’. All my attempts to use common dictionaries to disclose the sense of such usage of  this word were not successful. Probably some explanation can be inserted at the first use of the term.

Response 2: We thank the reviewer for pointing this out. We have changed the text accordingly to make it more understandable.

Point 3: Probably authors will re-consider the use of commercial names as ‘Tricalcii phosphas’ (page 6) in favor IUPAC names of chemical compounds – in this case ‘tricalciumphosphat’?

Response 3: We thank the reviewer for pointing this out, we made the changes in the text.